# Design, Implementation and Practical Evaluation of an Opportunistic Communications Protocol Based on Bluetooth Mesh and libp2p

**DOI:** 10.3390/s25041190

**Published:** 2025-02-15

**Authors:** Ángel Niebla-Montero, Iván Froiz-Míguez, Paula Fraga-Lamas, Tiago M. Fernández-Caramés

**Affiliations:** 1Department of Computer Engineering, Faculty of Computer Science, Universidade da Coruña, 15071 A Coruña, Spain; angel.niebla@udc.es (Á.N.-M.); ivan.froiz@udc.es (I.F.-M.); tiago.fernandez@udc.es (T.M.F.-C.); 2Centro de Investigación CITIC, Universidade da Coruña, 15071 A Coruña, Spain; 3Centro Mixto de Investigación UDC-Navantia, Universidade da Coruña, Edificio de Batallones, s/n, 15403 Ferrol, Spain

**Keywords:** opportunistic networks, opportunistic edge computing, OEC systems, opportunistic IoT, Bluetooth 5, decentralized IoT

## Abstract

The increasing proliferation of Internet of Things (IoT) devices has created a growing need for more efficient communication networks, especially in areas where continuous connectivity is unstable or unavailable. Opportunistic networks have emerged as a possible solution in such scenarios, allowing for intermittent and decentralized data sharing. This article presents a novel communication protocol that uses Bluetooth 5 and the libp2p framework to enable decentralized and opportunistic communications among IoT devices. The protocol provides dynamic peer discovery and decentralized management, resulting in a more flexible and robust IoT network infrastructure. The performance of the proposed architecture was evaluated through experiments in both controlled and industrial scenarios, with a particular emphasis on latency and on the impact of the presence of obstacles. The obtained results show that the protocol has the ability to improve data transfer in environments with limited connectivity, making it adequate for both urban and rural areas, as well as for challenging environments such as shipyards. Moreover, the presented findings conclude that the protocol works well in situations with minimal signal obstruction and short distances, like homes, where average latency values of about 8 s have been achieved with no losses. Furthermore, the protocol can also be used in industrial scenarios, even when metal obstacles increase signal attenuation, and over long distances, where average latency values of about 8.5 s have been obtained together with packet losses of less than 5%.

## 1. Introduction

In recent years, the number of Internet of Things (IoT) devices that require connectivity to function properly has grown, demanding the development of more efficient networks [1]. Opportunistic networks, which allow for the intermittent and temporary sharing of information among devices [2,3], have achieved increased importance and are considered a viable option for areas where continuous connectivity is not guaranteed and traditional network infrastructure may be compromised, such as rural areas [4], congested urban areas, or emergency situations such as natural disasters [5].

Opportunistic networks are a useful alternative when continuous access, whether to the Internet or to a Local Area Network (LAN), is missing [6]. These networks allow for effective data sharing and routing even in the lack of constant connectivity. They also help to overcome some of the constraints associated with Cloud Computing, such as scalability, lack of security mechanisms for resource-limited devices, and excessive energy consumption [2]. To enhance Cloud Computing, technologies such as Edge Computing have emerged [7]. When combined with opportunistic networks, they allow for creating Opportunistic Edge Computing (OEC) systems, which provide Edge Computing services in an opportunistic way [8,9].

Bluetooth stands out as one of the most used technologies in the field of IoT [10,11]. For instance, Bluetooth Low Energy (BLE) has been developed as one of the most energy-efficient wireless communication technologies, with a high transmission–energy ratio [12]. BLE Mesh allows for completely decentralized operation [13], with advertising for node communications, making it ideal for opportunistic communications. Despite these characteristics, it has certain limitations [14,15]:It requires continuous activity to maintain the mesh network, which usually involves increasing power usage.Messages have to go through numerous nodes before reaching their destination.Network congestion may occur in environments with high node density.The limited memory and processing resources of some nodes may lead to making it more difficult to manage the complexity of the network.Large volumes of high-speed data cannot be sent without latency limitations.

This article describes a communication protocol based on libp2p [16] and Bluetooth Mesh for two different home IoT and Industrial IoT (IIoT) scenarios. The developed protocol enables decentralized and opportunistic communication among nodes so that each node acts both as a client and a server, allowing for direct data exchange between them, as well as taking advantage of communication opportunities that arise in a dynamic and non-predefined way.

The following are the main contributions of this article:To the authors’ knowledge, this is the first article that demonstrates the use of Bluetooth 5 in combination with libp2p for implementing practical opportunistic networks. Libp2p is a modular networking library initially developed as part of the IPFS (InterPlanetary File System) project [17]. It provides a flexible architecture for developing P2P applications that manage tasks such as peer discovery, message routing, and connection management via various transport channels. Libp2p is a relevant option for P2P communication as it supports multiple network topologies and protocols, making it adaptable to different environments and resilient to network changes.It details a novel integration of libp2p with Bluetooth 5 in order to enable peer-to-peer (P2P) features like peer discovery and decentralized management, which enhance network flexibility and reliability.The proposed opportunistic developments are evaluated in real home and industrial scenarios, including inside a military shipyard workshop, evaluating latency and the impact of physical obstacles on network performance and providing key data on the effectiveness of the architecture.

The rest of this article is structured as follows. Section 2.3 analyzes the current state of the art about opportunistic communication networks. Section 3 describes the proposed communications architecture. Section 4 details the implementation of the architecture. Section 5 presents the performed experiments. Finally, Section 7 is devoted to conclusions.

## 2. Background Work

### 2.1. Previous Work

Opportunistic systems have been previously discussed in [2,18,19,20], where their essential parts are described. Specifically, such papers indicate that an OEC system is defined as a decentralized and adaptable architecture that allows IoT/IIoT devices to take advantage of computing, storage, and communication resources available in their nearby environment without relying on continuous connectivity or predefined infrastructure. Its operation is based on the ability to dynamically detect and use nearby nodes or gateways (static or mobile) to execute critical tasks in the absence of stable network coverage.

In order to clarify how an opportunistic system works, an example of a generic architecture is shown in Figure 1. At the bottom, the IoT network layer is composed of different IoT networks whose end devices are capable of exchanging data with the upper layer. Such end devices can also communicate with other devices in their same network, thus acting as relays. What distinguishes the IoT network layer from other layers present in generic IoT architectures is the fact that the communications with the upper layer are not always possible, so they need to be carried out opportunistically. Therefore, the upper layer (labeled in Figure 1 as OEC Smart Gateway Layer) is made up of gateways capable of providing services opportunistically with reduced latency, thanks to their proximity to the IoT end nodes. Moreover, the proposed architecture allows the received data to be stored in a distributed manner among all the gateways. Finally, the Cloud provides services that gateways cannot provide, such as intensive processing or the storage of large amounts of data.

An example of the use of the previously described communications architecture is detailed in [2]. Such a paper proposes an OEC system in which IoT devices deployed in remote areas or with limited connectivity (e.g., environmental or industrial sensors) use Bluetooth 5 to dynamically connect to edge gateways based on Single-Board Computers (SBCs) (e.g., Raspberry Pi) when these are within their communication range. Equipped with Bluetooth 5 modules and WiFi/4G connectivity, such gateways act as intermediary nodes that provide local storage and processing services, reducing the dependency on a continuous Internet connection. The system employs a peer-to-peer (P2P) network based on Kademlia DHT to store data on other nearby gateways or route them to available destinations, ensuring service continuity even when a node loses its connectivity and data locally. The communication between gateways is primarily performed via WiFi or 4G networks, but in the event of interruptions, the system turns to the Cloud as a backup to maintain connectivity between geographically dispersed IoT networks.

### 2.2. Disaster Relief Use Case

To illustrate the use of opportunistic systems, the example of a disaster relief scenario related to a flood is shown in Figure 2. In such a situation, the environment is rapidly affected by the flood, so quick decisions need to be made. To tackle these issues, it is assumed that there are IoT sensors deployed across the scenario to detect the water level. Such sensors monitor risk conditions characterized by high water levels or heavy rains and are connected to gateways that enable sending the collected information to servers located in a remote Cloud, where it is analyzed to send alerts. However, in the case of a flood, the connectivity of the deployed IoT water level sensors can be affected, making it not possible to send the data to the Cloud (step 1 in Figure 2). Nonetheless, when using an opportunistic system, devices like drones, wearables, or mobile phones can act opportunistically as mobile OEC Gateways to collect information from the sensors (step 2) or provide them with Edge Computing services. It is important to note that, in the event that one of the OEC gateways fails or when a connection cannot be established with an OEC gateway, it would be necessary to wait for another mobile node to appear opportunistically to reestablish the connection. Thus, once one OEC gateway collects the information, it is routed through the network infrastructure, being replicated among all the gateways (step 3) until reaching the one that can connect to the Cloud (step 4). The Cloud can then carry out the necessary actions, such as activating the flood alert system (step 5) or sending alerts to the remote users closest to the affected area (step 6) (these latter actions can also be performed in an opportunistic way, but, for the sake of clarity of Figure 2, they are represented as direct actions).

### 2.3. Analysis of the State of the Art

Opportunistic communication networks have become a viable option in recent decades for situations where standard connectivity is either nonexistent or severely constrained [21]. These systems make use of connection opportunities in dynamic and irregular connectivity environments by enabling direct communications between mobile devices and other nodes. As technology develops, a variety of applications in crucial fields like disaster relief [5], environmental monitoring [4], and people tracking [22] are made possible by the combination of opportunistic networks with cutting-edge technologies like IoT, mobile Edge Computing (MEC), or 6G networks.

In the past, some researchers have focused on the design and optimization of communication protocols and routing techniques for opportunistic networks. Specifically, some researchers analyzed how routing protocols and communication approaches might be improved to respond to the changing conditions of opportunistic networks. For instance, in [23], the authors describe a forwarding factor routing protocol that enhances data forwarding by dynamically choosing nodes based on their forwarding capabilities, which are influenced by various parameters including node mobility and network structure. Compared with static routing systems, the authors’ techniques improve data delivery rates and reduce latency, which makes them suitable for IoT scenarios with high mobility and fluctuating node density. Recent work on information freshness (Age of Information, AoI) optimization, such as [24], demonstrates how time-modulated arrays can prioritize time-sensitive data in covert communications, a concept relevant to opportunistic networks where minimizing AoI is critical for applications like disaster response or real-time monitoring. In addition, in [25], a technique is presented that considers node energy levels and connection stability to make a dynamic selection of forwarding nodes, decreasing energy usage, extending network useful life, and assuring reliable data transmission. Another interesting work is described in [4], where a fuzzy Q-learning algorithm is applied to vehicular crowdsensing networks, enhanced by MEC. Integrating fuzzy logic with Q-learning allows for more sophisticated routing and data dissemination decisions, taking into account elements such as vehicle speed, direction, and network connectivity. Finally, it is also worth mentioning that there are emerging frameworks for integrated sensing and communications, like the tensor-based approach described in [26], which unifies channel estimation and target detection in massive MIMO systems, providing insights into how opportunistic networks could dynamically adapt RF parameters (e.g., transmit power, beamforming) to improve reliability in unstable environments.

Other articles examine the integration of opportunistic communication systems with advanced technologies such as 6G, MEC, or blockchain to enhance their capabilities and applications. For example, in [5], the authors utilize 6G networks to enhance the responsiveness and effectiveness of rescue operations for drone-based disaster response. In [27], blockchain technology is integrated into IoT networks using opportunistic block validation to enhance security and trust, utilizing the decentralized and immutable nature of blockchain. Moreover, in [28], the authors explore how opportunistic networking principles can be integrated into IoT systems for smart city applications to improve connectivity and data sharing. Such a study highlights the potential of opportunistic communications to improve network scalability, to reduce infrastructure costs, and to enhance the overall performance of IoT deployments.

Other articles provide comprehensive evaluations, highlighting difficulties, significance, and prospective applications for opportunistic networks. For instance, in [29], important challenges such as irregular connectivity and the need for effective routing strategies are discussed. Such a paper also emphasizes the importance of opportunistic networks in situations where traditional networks are unavailable or overloaded, such as in disaster zones or densely populated urban regions. Despite substantial progress, the paper indicates that universal deployment has not been achieved and recommends continued research into new use cases and integration with emerging technologies such as 5G and IoT. Related to the previous article, in [30] the authors utilize wireless network identifiers (beacons) for information dissemination in opportunistic networks. This solution makes use of existing wireless network infrastructure, such as Wi-Fi or BLE, to transmit messages without requiring active connections. It is especially useful in situations requiring quick information circulation, such as emergency notifications or public announcements.

With respect to Bluetooth, research has been carried out on how to apply and optimize BLE for opportunistic networking, with an emphasis on data dissemination and energy efficiency. For instance, a mesh network protocol using BLE technology is presented in [31] and is aimed at smart homes, IoT, and industrial automation. It makes use of the publish/subscribe paradigm, which enables nodes to accept data from publishers and to subscribe to certain types of messages. Other authors have made use of BLE for opportunistic networking, as in [32], which describes techniques to improve data dissemination and to decrease energy consumption by making effective use of BLE advertising and scanning modes. Similarly to opportunistic networks, the proposed protocol ensures efficient data exchange even in situations with sporadic connections.

After reviewing the current state of the art, it can be concluded that the previously mentioned studies have been mostly focused on simulations and on the theoretical part, providing barely any experimental evaluation in real-world scenarios. Moreover, few fully address security and privacy concerns, and there is little emphasis on integrating emerging technologies like Bluetooth Mesh with opportunistic networks.

In contrast, this article provides both a theoretical description and a practical validation in real scenarios, including specific details on security, privacy, and novel strategies to optimize power consumption in BLE devices. Furthermore, this work also focuses on making opportunistic networks more robust and adaptable to changing conditions, such as poor coverage situations, which fills an important gap in the existing literature.

## 3. Design

Figure 3 shows the proposed OEC IoT architecture for an IIoT environment, focusing on pallet asset management, autonomous vehicles, and wearables. Such an architecture is composed of three main layers:IoT Network Layer: this layer contains IoT nodes deployed at different spots in a shipyard. These nodes are connected to elements such as pallets, autonomous vehicles, and wearables, allowing them to monitor and exchange relevant data among them. Pallets can be equipped with sensors that allow for the identification and monitoring of their position and/or status. Autonomous vehicles are connected to the IoT network to facilitate the collection and transportation of goods, possibly with constant communication to receive route instructions or real-time updates. Wearables are carried by operators and are used for activity tracking or for monitoring their status and location.OEC Smart Gateway Layer: this layer consists of OEC gateways that serve as intermediaries between the IoT nodes in the local network and the Cloud. Each gateway is responsible for aggregating and processing the information coming from the different IoT nodes in the lower layer and for transmitting such information to other gateways or to the Cloud if necessary. Gateways communicate with each other, allowing data to flow dynamically and flexibly. This communication between gateways enables creating a distributed network that can operate even if part of the network loses its connectivity. In order to ensure that the functionality of the Gateways is not affected, it is necessary to assume that they have sufficient resources, such as a stable power supply and a sufficiently high processing capacity.Cloud layer: this layer provides a centralized platform for managing large amounts of data, long-term storage solutions, and long-distance communications that may not be possible through edge-based processing.

To develop the previously described architecture, the protocol stack shown in Figure 4 was devised. The lower protocols of such a stack manage data transmission and linking, while the Opportunistic Network Layer (OEC Network Layer) handles critical discovery and routing services in the distributed network. The Application Layer employs such services to provide functionality and value to users and end devices. Specifically, such layers operate as follows:The Physical (PHY) Layer is in charge of sending data at the hardware level, via physical media such as radio waves or any wireless or wired communications technique.The Link Layer manages the data link and assures packet transfer between nodes that are directly linked to one another. Error detection and node synchronization are handled here.The Network Layer enables the movement of data packets between devices that are not physically linked. Thus, it defines how data are transported across network nodes.The Transport Bridge acts as an intermediate layer between the lower layers and the Opportunistic Network Layer (OEC). It is responsible for facilitating interoperability between the protocols and different technologies at the upper and lower layers.The OEC Network Layer is dedicated to managing the characteristics of opportunistic networks. It includes three key services:–Peer discovery: in charge of finding nearby nodes or devices, allowing opportunistic nodes to connect when they are in communication range.–Peer routing: manages how data should be routed between peers (nearby devices). Thus, it selects the best routes based on node availability and capacity.–Data routing: similar to peer routing, but focused on managing the effective transfer of data between network nodes, ensuring that information reaches its destination efficiently, even with the presence of intermittent links.Finally, the Application Layer is responsible for displaying data to end users or systems and for providing the interface for users to interact with the IoT network.

## 4. Implementation

### 4.1. Communications Architecture

The proposed communications architecture, which is based on a modular structure with mobile nodes and readers interconnected through Bridges, uses a combination of transport (Bluetooth 5 [33]) and a network protocol (libp2p) to handle communications. Among the potential available communications technologies, Bluetooth 5 is preferred over other technologies like Wi-Fi, LoRa [34], or Zigbee [35] because, as it can be observed in Table 1, it provides a better balance among power consumption, data speed, range, and mesh networking capabilities. This is especially true for mobile-centric applications where seamless connectivity with smartphones and wearables and energy efficiency are critical.

Figure 5 shows the protocol stack with the protocols selected for the implementation. In addition, libp2p has been chosen over other P2P protocols like IPFS or ZeroMQ [36] because of its flexibility, modularity, and integrated support for peer discovery, routing, and security. It can easily incorporate several transport protocols, including Bluetooth, TCP, WebRTC, and others, which enables it to adjust to the changing conditions and limited resources that opportunistic communication networks frequently require.

The OEC Network Layer plays an important role in managing P2P communication, combining discovery, routing, and security strategies:FloodSub: used for opportunistic routing between nodes, allowing messages to be relayed on the network without the need for a predefined hierarchical structure. The protocol is explained in more detail in Section 4.5.3.BLE advertising: uses Bluetooth 5 advertising capabilities to allow nodes to advertise their presence on the network without requiring active connections. This allows nodes to discover new opportunistic peers before establishing persistent connections.Yamux [37]: a multiplexing protocol that allows multiple data streams over a single underlying connection. It is employed within libp2p. It facilitates efficient management of multiple peer-to-peer communication sessions without opening new connections.Noise [38]: a lightweight cryptographic framework used in protocols such as libp2p to provide authentication and encryption. It ensures the confidentiality and integrity of messages exchanged between nodes.

Figure 6 depicts the implemented communications architecture, which, for the sake of clarity, only contains two nodes: one represents a mobile node while the other is a reader (i.e., a fixed node). Each node has a libp2p host and a Bluetooth interface. Each host has a transport layer that is implemented thanks to a bridge that interacts over the Bluetooth interface.

### 4.2. Communications Protocol

The sequence diagram shown in Figure 7 depicts the host construction procedure and how messages are handled during the initial interaction between two opportunistic nodes. A thorough description of the protocol sequence is provided below:First, NodeA initializes the Bridge (step 1A), which opens a port to enable communications (step 1B).The libp2p host is initialized (step 2A) and returns a peer ID (step 2B), which is a unique identifier assigned to each node in the network, allowing for identification and communication between peers.NodeA configures an event handler (step 3A) while the Bridge generates a stream for performing bidirectional communications (step 3B).NodeA is initialized by setting up its publish/subscribe (Pub/Sub) system (step 4), which facilitates a messaging scheme in which publishers send messages to subscribers that do not know each other, allowing for detached communication in distributed networks.NodeA attempts to connect to NodeB (which has previously performed similar steps to NodeA, but which have not been included in Figure 7 for the sake of clarity) by sending the previously generated peer ID through the Bridge (steps 5A and 5B). The Bridge receives the peer ID from NodeB (step 5C) and delivers it to NodeA (step 5D).NodeA starts the connection with NodeB by sending a connection initiation request to the libp2p host (step 6).The libp2p host initiates the connection to NodeB (step 7A) and communicates through the Bridge to attempt to establish a connection (step 7B). The Bridge receives the response (step 7C) and forwards it to the libp2p host (step 7D).The peers negotiate the multistream protocol to be used in the same way as negotiated in step 7. Such a protocol guarantees that both nodes use the same protocol to exchange information after connecting.The peers negotiate the encryption protocol in the same way as negotiated in step 7. Before exchanging any data, both nodes agree on how to encrypt messages (for example, TLS or Noise). This ensures that the transmitted data are not intercepted or manipulated by third parties.The peers negotiate the multiplexer protocol in the same way as negotiated in step 7. This protocol enables multiple data streams to be sent over a single connection, allowing several operations (such as data transfer and control messages) to be performed simultaneously without the need for establishing new connections.The peers negotiate the Pub/Sub protocol in the same way as negotiated in step 7. This is used for propagating messages across the network without requiring a node to communicate directly with another specific node.

### 4.3. Bridge

The Bridge interconnects libp2p with the Bluetooth transport layer. In order to perform such an interconnection, it creates a process that listens and writes to the serial port; in this way, the messages required for the libp2p connection travel over the Bridge and are transmitted to the Bluetooth module via the serial port.

#### 4.3.1. Data Structure

The Bridge processes libp2p communications, whose messages are encoded as UTF-8 byte strings; therefore, to be delivered via serial port, they must be first converted to hexadecimal and then into a series of bytes, appending a start and an end element. Moreover, the messages range in size from 20 to 2000 bytes, but the proposed Bluetooth transportation layer can only send 255 bytes, so messages need to be split in segments. To avoid saturating the channel, each segment is queued and broadcast with a specific delay. In addition to the message start and end characters, each segment contains a header that indicates the entire size of the unfragmented message, which is eventually processed by the receiver. Table 2 illustrates the internal structure of the frame, which contains the following fields:Header (13 bytes): it contains the message header, which includes the information necessary to send the message through the OEC protocol.Dst (6 bytes): it contains the destination ID, which identifies the node to which the message is addressed.Start Char (2 bytes): it indicates the start of the message character, which serves as a delimiter so that the receiver knows where the message content begins.Data (1–227 bytes): payload of the message.Length (4 bytes): actual length of the message so that reception can determine whether the message has arrived complete.End Char (3 bytes): it indicates the end of the message character, which serves as a delimiter so that the receiver knows where the frame ends.

#### 4.3.2. Data Processing

The Bridge processes all the messages that arrive over the serial connection. If the Bridge recognizes an incoming message with the appropriate start character, it begins processing the message; if the message does not have an end character, it indicates that the message is a segment, and it must continue reading the serial port until receiving the whole message.

After all segments have been received, the Bridge examines the length field to guarantee that the message was received completely and that no fragments were lost. It is not necessary to verify the integrity of the data at this point since this verification is performed at another level of the proposed opportunistic stack; specifically, libp2p is responsible for this verification once the data are decoded. The full message is then decoded from hexadecimal to UTF-8 and delivered to the function that handles libp2p communications.

### 4.4. Bluetooth Transport Layer

The Bluetooth networking protocol depends on the Bluetooth Transport Layer to provide communications between the mobile nodes and readers that serve as Bluetooth Mesh Clients. Large-scale networks may benefit greatly from the wireless mesh topology that a Bluetooth Mesh network offers, as it allows various devices or nodes to connect and to interact. Using the mentioned mesh clients makes it easier to discover peers without needing a central node to which all nodes must be connected to route messages. In this case, to find a peer, the node only has to send a broadcast message so that the rest of the clients provisioned in the same network know that they are available for the connection. In this way, they can start the connection, and the relationship between the peer ID and the mesh client ID will be stored in a table. For the rest of the messages, private and encrypted messages will be used between the two nodes.

### 4.5. Libp2p Host

To mitigate the weaknesses of Bluetooth Mesh at the network level, it is possible to make use of P2P solutions. For instance, the integration of libp2p with Bluetooth Mesh provides significant advantages, especially when it comes to opportunistic communications and lowering the inherent limitations of Bluetooth Mesh. The following are the benefits of using libp2p in conjunction with Bluetooth Mesh to address the limitations of the technology:Mesh networks are limited to short-range communications, so while Bluetooth Mesh would handle local broadcast, libp2p would handle a broader broadcast to the rest of the nodes.Mesh networks are not designed to withstand sudden disconnections; in an opportunistic network, this is essential so libp2p can help data persist on the network by allowing message propagation even if the source nodes go offline.If by any chance the Bluetooth transport is limited, Libp2p acts as a higher level of connection management, allowing interoperability between different transport technologies (not only Bluetooth Mesh, but also Wi-Fi or LTE, if available).Although Bluetooth Mesh is designed for low-power nodes, message retransmission within the mesh network can be somewhat inefficient due to continuous retransmissions, libp2p can optimize global routing and avoid redundant retransmissions, preventing messages from propagating further than necessary.

In addition, other reasons to use libp2p in opportunistic applications include the following:The networking stack may be customized to fulfill unique requirements thanks to its modularity.It provides a collection of standards that may be tailored to support a variety of transport protocols.It offers a range of discovery mechanisms and data storage.It contains multiple security features such as peer identity verification using public key cryptography and encrypted communications between peers using modern cryptographic algorithms.It includes features such as peer discovery and content routing that assist in guaranteeing that the network remains available and accessible even if some peers are offline or unreachable.It uses publish/subscribe messaging (Pub/Sub) to send a message to several receivers without the sender knowing who they are.The decentralized design enables the applications to function without a central authority.

#### 4.5.1. Negotiation

Messages are delivered as UTF-8 byte strings, which are always followed by a newline character. In addition, each message is prefixed with its length in bytes (including the new line), which is encoded as an unsigned variable-length integer by the multi-format unsigned variant standard.

For example, the text “/multistream/1.0.0” is delivered as the hexadecimal value 0x032F6D756C746973747265616D2F312E302E300A. The first byte is the variant-encoded length (0×03), followed by the short name for the multistream selection protocol “/multistream” (0×2F 0×6D 0×75 0×6C 0×74 0×69 0×73 0×74 0×72 0×65 0×61 0×6D) and a version number “/1.0.0” (0×2F 0×31 0×2E 0×30 0×2E 0×30), and finally the new line (0×0A).

Multistream selection is a mechanism used in libp2p to handle peer connections. Its main purpose is to facilitate protocol negotiation over an existing connection in an efficient and flexible way. Such a mechanism is based on an exchange of messages using text strings to identify the available protocols. Each peer sends a message containing the name of the protocol it wishes to use. If the other peer supports that protocol, it responds with a confirmation.

Figure 8 depicts the fundamental sequence for the “multistream-select” negotiation. The negotiation for establishing a connection between two nodes is as follows:The initiating peer establishes a channel with the destination peer; this channel could be a new connection or a new stream multiplexed over an existing connection.Both peers will then broadcast the multistream protocol ID to indicate that they want to employ multistream selection. Both sides can deliver the first multistream protocol ID without waiting for data from the other side. If either side obtains anything other than the multistream protocol ID as the initial message, the negotiation process terminates.After both peers have agreed to employ multistream selection, the initiator communicates the protocol ID they want to use (step 3A). If the recipient supports the protocol, it will answer by repeating the protocol ID to indicate its agreement (step 3B). If the protocol is not supported, the recipient will respond with the string “na”, indicating that the requested protocol is unavailable (step 3C).If the peers agree on a protocol, the task of selecting numerous streams is completed, and future communication across the channel will follow the agreed-upon protocol rules. If a peer receives a “na” answer to a proposed protocol ID, it may retry with an alternative protocol ID or terminate the channel.

#### 4.5.2. Upgrading Connections

Since there are several ways to provision libp2p capabilities, the connection update process uses protocol negotiation, as described in the preceding section, to choose which specific protocols to use for each capability. The protocol negotiation procedure employs multistream selection. When a connection requires both security and multiplexing, a secure communication channel is first established, and stream multiplexing is negotiated via an encrypted channel.

Figure 9 shows an example of the connection updating procedure. The following are the main steps:Both peers provide the multistream protocol identifier, indicating that they will utilize multistream-select to negotiate protocols for the connection update.The starting peer recommends the TLS protocol [39] for encryption, which the destination peer rejects since it does not support TLS.The initiating peer then offers the Noise protocol [38], which is accepted by the destination peer, who acknowledges it by returning the Noise protocol identification.At this point, the Noise protocol takes over, and the peers perform the Noise handshake to create a secure channel. If the Noise handshake fails, the connection formation procedure terminates. If successful, the peers will utilize the safe channel for all further messages, including the remainder of the connection update procedure.Once the secure communications channel has been established, peers decide the stream multiplexer to be utilized. The negotiation method is the same as the previously described, with the calling peer offering a multiplexer and delivering its protocol identification, and the listening peer answering by returning the admitted identifier or sending “na” if the multiplexer is not compatible.Once both security and stream multiplexing are established, the connection update procedure is complete, and both peers can utilize the resultant libp2p connection to create new secure multiplexed streams.

#### 4.5.3. Publish/Subscribe Mechanism

Pub/Sub [40] is a communications model that allows peers to exchange messages about topics of interest. The subscription to a topic means that the subscribed nodes will receive the messages related to it. Peers can send messages to topics, and each message is distributed to all peers that have subscribed to that topic. Before a peer can subscribe to a subject, it must first locate and connect with other peers via the network. Pub/Sub does not have a built-in method for discovering peers.

FloodSub [41] was utilized for the development presented in this article. It is a basic implementation of the Pub/Sub paradigm for P2P networks that is intended for usage in decentralized contexts. In FloodSub, all nodes in the network serve as both publishers and subscribers, which means that any node may post messages and subscribe to topics.

When a node subscribes to a topic, it indicates such a subscription to its neighboring nodes. When a node publishes a message on a subject, all of its neighbors obtain it. These neighbors, in turn, spread the word to their neighbors, and so on. This procedure continues until the word has spread across the whole network.

FloodSub has the benefit of not having a central server, making it ideal for P2P networks in which all nodes are equal. However, this strategy has drawbacks since it might flood the network and consume excessive bandwidth, reducing the scalability of the protocol.

#### 4.5.4. Inherent Challenges

Libp2p faces some challenges when used in opportunistic networks, mainly due to the dynamism of these topologies. Key among them is adaptability to perpetual changes in the network; the rapid entry and exit of nodes makes it difficult to maintain effective connections and routes, often increasing the time and resources needed to reestablish them when they are lost. Furthermore, peer discovery, which is crucial in such networks, can be very expensive in terms of bandwidth and energy, especially in dense or heterogeneous networks where nodes use different transport technologies.

Routing challenges also present significant challenges; libp2p algorithms may not be fast and efficient enough in environments where routes change too frequently, potentially introducing latency or redundancy in message propagation. Resource consumption is another major limitation, as libp2p can be demanding in terms of energy and computational capacity, especially on IoT devices that have limited batteries and processing capabilities.

Due to these limitations, it has been decided to delegate peer discovery to Bluetooth Mesh, while libp2p would be responsible for managing global connectivity and routing in the network. In this way, once a peer is discovered, libp2p would store the Bluetooth Mesh ID in its routing table, as shown in Figure 6.

### 4.6. Bluetooth Mesh Fine-Tuning Segmented Messages

The BLE Mesh protocol that makes use of the OEC protocol described in this article internally exhibits a ’flooding’ behavior in its communications because it does not use routing tables. This makes it extremely efficient when transmitting small messages, but, because the designed protocol uses messages with a high payload to use libp2p, improvements need to be performed by making certain internal adjustments to the BLE Mesh protocol [42].

First, it must be noted that the transmission of a single segment depends on the number of retransmissions and on the initial interval between retransmissions. Specifically, the average transmission of N segments can be determined using the following formula:(1)$$\begin{array}{c} T_{avg}=\left(10+\left(T_{int}+10\right)*T_{count}\right)*N \end{array}$$

*T_count_* determines the amount of extra transmissions for a certain message; if set to zero, it implies transmitting a single message, which implies sending a total of three packets (one for each advertisement channel), as long as the message does not need segmentation (i.e., it has a length of 11 bytes or less). *T_int_* limits the duration in milliseconds of the sending interval of these extra messages, and *N* is the number of segments.

Using 0 additional transmissions would result in the lowest message sending time of 10 ms. However, because segmented messages are sent in a maximum of 32 segments, the maximum transmission would be 320 ms if no additional messages were used. Moreover, for segmented messages, the transport layer must also be considered. It is important to note that segmented messages, unlike unsegmented messages, must always employ ACKs.

Fortunately, Mesh version 1.1 allows developers to fine-tune segmented messages using several settings of the Segmentation and Reassembly (SAR) protocol used at the transport layer. One of the most important settings is the duration of the gap between two successive segments in segmented messages (both transmitting and receiving), which is determined by the following two formulas:(2)$$\begin{array}{c} Tx_{seg\_int}=\left(Tx_{seg\_int\_step}+1\right)*10\\ Rx_{seg\_int}=\left(Rx_{seg\_int\_step}+1\right)*10 \end{array}$$

*Tx_seg_int_step_* controls the interval between sending two consecutive segments in a segmented message, and *Rx_seg_int_step_* defines the segment reception interval step used for delaying the transmission of an acknowledgment message after receiving a new segment. By default, both variables are equal to 5 ms, implying a period of 60 ms between segment transmission/reception. When applied to a maximum of 32 segments, this corresponds to a time of 1920 ms, assuming that no further messages are utilized in the communication. However, by default, the number of additional resends is two, so the default periods would be significantly longer. These times greater than one second are quite similar to what is obtained when message sending is decoupled from the designed protocol, though buffers and caches are used to reduce the number of performed transmissions, so these times are not exact, but they provide an approximation. In this case the forwarding factor is more critical, since all the segments that make up the messages are forwarded.

Taking into account the opportunistic nature of some nodes, they could benefit from adjusting the previously described times in order to minimize the use of the radio module and thus consumption, as a constant traffic flow is not necessary. However, it is necessary to determine an optimal adjustment for the previously mentioned time parameters, since a reduction in the intervals can lead to collisions and to the need for retransmitting the segment (by default, one retransmission is used for unicast addresses and two for multicast). In opportunistic environments with low data traffic, a minor change to these intervals can result in significant time improvements.

## 5. Experiments

The tests presented in the next sections were designed to validate the functionality of the proposed protocol in a controlled static environment, such as the inside of a home, and in an industrial environment, such as the interior of shipyard workhouses. The showcased use cases involved the exchange of messages between two nodes, with one acting as the sender and the other one as the receiver. Specifically, during these experiments, we have focused on latency measurement. The choice of latency as a priority metric is based on its direct impact on the effectiveness of critical IoT applications, where data must reach their destination within narrow time windows to trigger timely responses. In scenarios such as disaster response, the timing of data reception plays an important role in the decision-making process. In opportunistic networks that allow messages to be sent even when there is a temporary disconnection and where the arrangement of nodes changes frequently, it is crucial to minimize latency so that there is enough time for the message to reach its destination before the connection times out. For the first experiment, a total of four tests were carried out, each with 100 packets transmitted with a transmission time of 9 s between each message. The objective was to see how different locations impacted latency and the number of lost packets. All source code needed to replicate the presented experiments is available online [43], allowing any researcher to use and further extend the developed OEC system.

### 5.1. Experimental Testbed

An experimental testbed was built to assess the performance of the proposed OEC system. The elements of such a testbed were two nodes built on a Raspberry Pi 3B single-board computer (SBC) with a Nordic nRF52840 development kit [44] linked via serial port to enable Bluetooth Mesh functionality. Figure 10 depicts one of such nodes. After the developed software was flashed onto the nodes, the Nordic nRF Mesh app for Android was used to provision and configure the nodes. Figure 11 shows two screenshots of nodes provisioned through the app. Two nodes that have already been provisioned are shown on the left (“0×0027” and “0×0023”), and the configuration of one of them, on the right, where a key has been linked and the publish and subscribe settings have been set up. This experiment was carried out by creating the group “0×C000”, where each node is configured to both publish and subscribe, enabling only the nodes that are subscribed to that group to view the content that they publish.

The Bluetooth Mesh provisioning controller handles provisioning for Nordic development kits. It supports four alternative out-of-band (OOB) authentication methods and uses the Hardware Information driver to generate a deterministic UUID that identifies the device.

### 5.2. Home Automation Experiments

The first set of tests was carried out in a static indoor environment, as shown in Figure 12, which represents the interior of a home, where there were typical barriers, such as walls and furniture. The distances between the deployed nodes were receiving (4–6 m). Figure 12 shows on a floor plan the four different positions where the receiving node (in red) was deployed around the environment and only one position for the sender (in green), which was utilized to send data packets to each node.

#### Communications Latency

Figure 13, Figure 14, Figure 15 and Figure 16 show the communications latency of messages transmitted using the developed protocol at 9 s intervals from four distinct places in a house: the kitchen, the entrance, the living room, and the bedroom. On the one hand, Figure 13 and Figure 14 show slightly more stable behavior than for the other two locations. It is worth noting that in all the performed experiments, there is a peak around packet 60, indicated by vertical red lines in Figure 13. After checking the connection logs and ruling out possible connectivity failures, it can be confirmed that this peak occurs because after a certain number of minutes, the protocol verifies whether the communication is still active, which causes a delay in the remaining packets. Notwithstanding this anomaly, there are hardly any spikes in latency (i.e., latency variability is low). The typical delay is approximately 8 to 8.5 s. This suggests that both locations benefit from better signal propagation conditions, possibly due to fewer obstacles and shorter distances to the transmission point. On the other hand, the results shown in Figure 15 and Figure 16 show less homogeneity in latency, with more spikes. Latency readings typically range from 8 to 9 s, indicating that higher variability could result from increased attenuation caused by a greater number of obstacles or longer distances to the transmission point.

Figure 17 depicts the standard deviation of latency for the four different locations of the receiving node, showing how latency differs in terms of consistency between environments. When comparing the results shown in Figure 17 with the ones shown in Figure 13, Figure 14, Figure 15 and Figure 16, a huge spike can be observed on packet 60 that indicates the exception shown on the previous latency graphs. If such a spike is ignored, it can be concluded that the latency stability is slightly better for locations 1 (living room) and 3 (entrance), whereas oscillations are greater for locations 2 (kitchen) and 4 (bedroom). These locations show higher variability, likely caused by increased signal attenuation, the presence of more obstacles, or less favorable propagation conditions in those areas.

### 5.3. Experiments in an Industrial Scenario

The second set of tests was carried out in a static indoor environment, as shown in Figure 18, which represents the interior of a shipyard workshop, where there were many more obstacles than in the first experiment, including multiple pallets stacked with pipes on top, industrial machinery, gantry cranes, workers, etc. The distance between the transmitting node and the receiving node was 43 m for the first scenario and then 117 m for the second. Figure 18 depicts the deployment of two nodes (in red) and a single sender (in green) that sent data packets to each node. Figure 19 shows an image taken during the tests of the real position of the sender in the workshop, whereas Figure 20 shows the receiving node during one of the tests.

#### 5.3.1. Communications Latency

Figure 21 and Figure 22 show the communication latency of messages sent using the developed protocol at 9 s intervals from the two scenarios selected for the shipyard workshop. Figure 21 shows that latency remained more stable for the first scenario (i.e., with nodes at a distance of 43 m). In fact, latency variability is small, losing only one packet. The typical delay oscillates between 8 and 9 s. Figure 22 shows more variation than the second scenario (i.e., for a distance between nodes of 117 m) suffered more latency oscillation and spikes. In such a second scenario, latency usually ranged between 8 and 9 s, with peaks of up to 11 s and a total of five packets lost, which are related to the increase in the communications distance and to the existence of more obstacles in the signal propagation path inside the workshop. As with the home automation tests, a peak occurs (in this case, around packet 50). As it was previously mentioned, such a spike is due to the fact that, after a certain number of minutes, the protocol checks whether the communication is still active, causing a delay in the remaining packets.

#### 5.3.2. Packet Delivery Ratio (PDR)

In order to evaluate the reliability and adaptability of the protocol to different environments, the packets sent and received during 100 packet transmissions between two nodes (without including potential retransmissions) are compared in Figure 23. For the indoor environment, the protocol achieved a 100% PDR in the first three positions (i.e., all packets were delivered successfully) and 99% in the fourth one, thus demonstrating a solid performance in stable short-range scenarios. For the industrial environment, the obtained results were decisive in clarifying whether the protocol is actually valid for large industrial scenarios. The result obtained for the first position at 40 m was similar to the one for the fourth indoor spot, with a 99% PDR. However, for the second tested spot, where the distance exceeded 100 m, PDR decreased to 95%, which is expected, since being in a Non-Line-of-Sight (NLOS) industrial area, the longer the distance, the higher the signal propagation loss, and the larger the number of obstacles that can be found on the communications path between the emitter and the transceiver. In any case, despite this 5% loss in the PDR (which, for instance, could be reduced by adding more retransmissions to the protocol), it can be concluded that the proposed system can be used in industrial environments similar to the ones tested (i.e., with long distances and with the presence of multiple large metallic obstacles).

## 6. Key Findings

After developing the proposed system and evaluating its basic performance, the following key findings can be provided:Opportunistic communications are an emerging field that has gained relevance for scenarios with restricted connectivity. Most of the previous research cited in this article focused on more theoretical approaches such as routing optimization or protocol efficiency for different network topologies, but there are not many practical evaluations. This article compensates for this lack by presenting a new multi-layer approach that combines Bluetooth Mesh for discovery and libp2p for communications, which improves adaptability and robustness in multiple environments.Use of other communications technologies. The protocol proposed in this article was designed to be agnostic to the physical layer and adaptable to multiple communication technologies, thus facilitating integration with emerging networks such as 6G or already well-established technologies like 5G, Zigbee, Wi-Fi, or LoRa, which were considered as potential alternatives during the design phase and can be used in future developments to create comparisons with the results obtained in this article for Bluetooth 5.Use of other application-level technologies. The devised protocol can incorporate at the application level technologies like blockchain, which can improve data integrity and decentralized management through the use of smart contracts or the storage of transaction records on the blockchain for audits and traceability [45].The results obtained during the experiments demonstrated that the protocol is robust against metallic obstacles and signal attenuation at over 100 m and maintains low latency in all scenarios, validating its suitability for industrial environments.Although the protocol was verified in domestic and industrial environments, it can be applied to dynamic scenarios such as vehicular networks (e.g., real-time traffic updates), natural disaster situations (e.g., drone swarms with intermittent connectivity), or smart cities (such as mobile sensor networks).Scalability. The performed experiments demonstrated the viability of using the system on the selected home and industrial scenarios, showing its basic functionality with a reduced minimal P2P environment with only two nodes. Such evaluations were carried out for this article to control all the variables of the experiments and thus accurately show the basic performance of the protocol, such as delivery latency and success rate, without interference from other factors. However, in real-world scenarios, Pub/Sub protocols are designed to operate in distributed networks with hundreds or thousands of nodes, so further research is still necessary in terms of scalability. Nonetheless, it is worth mentioning that some authors performed similar experiments with different protocols and technologies. For instance, in [46], the authors performed several experiments with 1000 nodes, a maximum mesh size of 12, and a round-trip time of 100 ms, achieving a 100% PDR when a Sybil attack [47] took place. Thus, the authors, by combining low latency, high resistance to attacks, and self-recovery mechanisms, were able to validate that their protocol was suitable for large-scale networks.

## 7. Conclusions

This article described a novel communication protocol that combines multiple technologies to enable decentralized and opportunistic communications among IoT devices, testing the performance and reliability of the proposed protocol by conducting tests that focused on latency and packet loss in various indoor and outdoor environments.

The results showed that the protocol works effectively in environments with minimal signal obstruction, maintaining constant latency and a reduced jitter, as in the home experiments. However, in other locations like industrial environments, with more barriers between the transmitting and receiving nodes, the performance was less stable, with higher latency variability and occasional spikes and packet drops. These findings imply that physical barriers and obstacles, such as large metallic objects, walls, and furniture, have a significant impact on the performance of the opportunistic protocol, particularly on signal strength and reliability.

Consequently, this paper demonstrated that the proposed protocol is a viable solution for implementing opportunistic systems based on Bluetooth 5 and libp2p in both industrial and home automation environments. The results highlight that the protocol achieves low latency, with minimum values around 8 s (depending on the environment). In addition, it can be stated that the protocol performs well in terms of reliability, with packet reception rates between 95% and 99% in the evaluated scenarios. These findings suggest that the protocol can be effectively deployed in diverse scenarios, although its performance can vary depending on environmental factors. Thus, although this article provides clear guidelines on how to make use of the developed OEC communications protocol (which is available as an open-source project), future researchers may need to optimize its operation by adapting it to the specific conditions of each application, including adjustments to the network topology or the incorporation of fault-tolerance mechanisms.

## Figures and Tables

**Figure 1 sensors-25-01190-f001:**
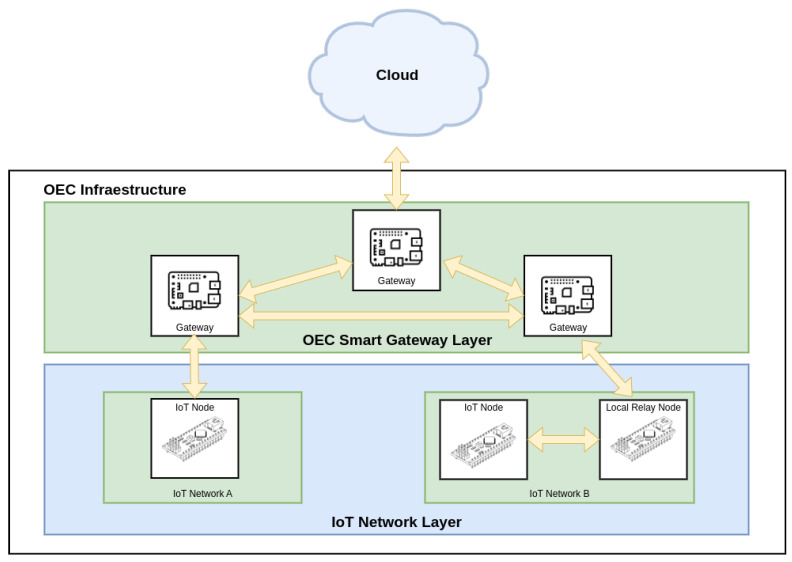
Generic OEC communications architecture.

**Figure 2 sensors-25-01190-f002:**
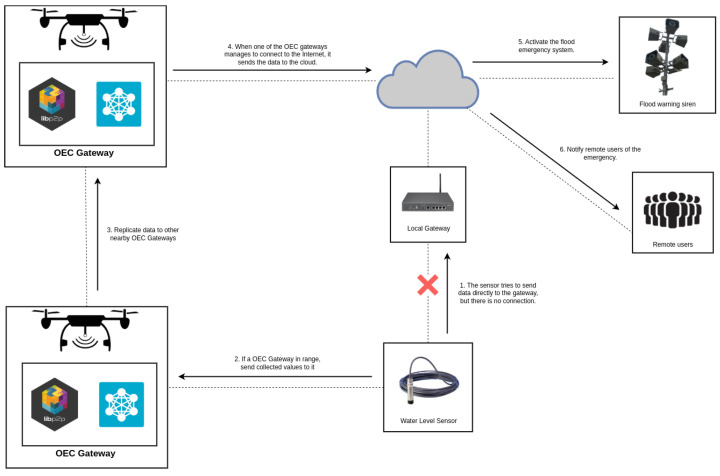
OEC in a disaster relief scenario.

**Figure 3 sensors-25-01190-f003:**
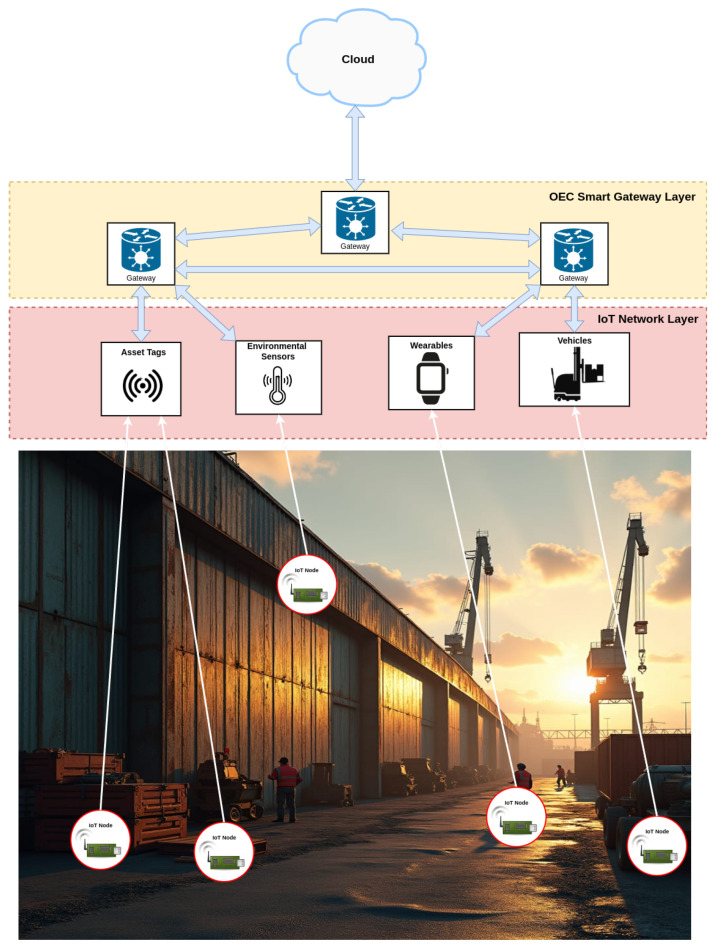
Proposed opportunistic communications architecture.

**Figure 4 sensors-25-01190-f004:**
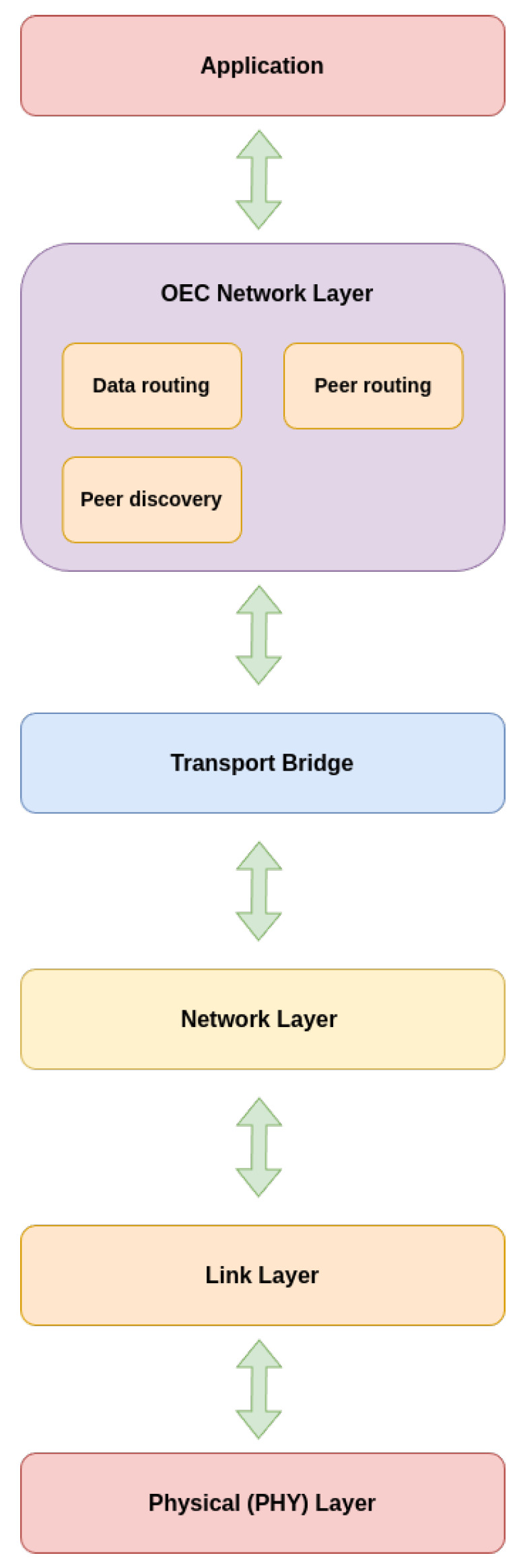
Proposed opportunistic protocol stack.

**Figure 5 sensors-25-01190-f005:**
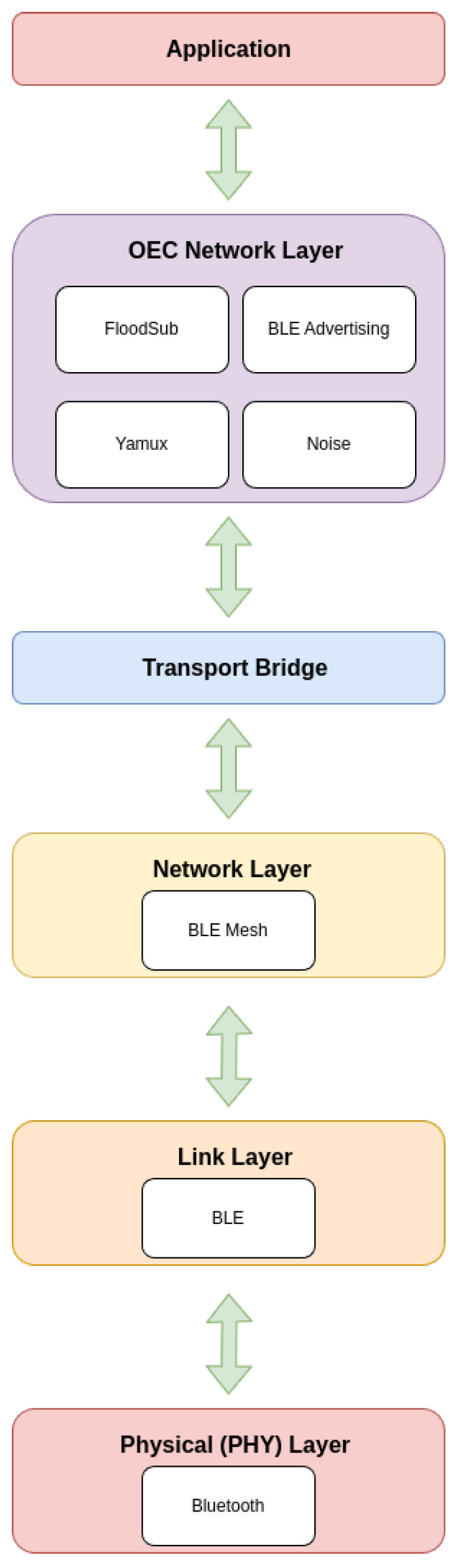
Implemented Bluetooth 5 and libp2p-based opportunistic protocol stack.

**Figure 6 sensors-25-01190-f006:**
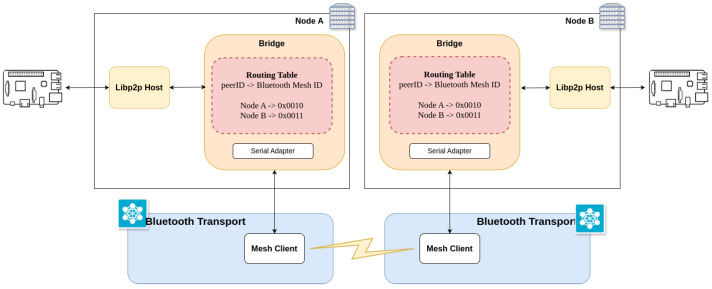
Implemented OEC communications architecture.

**Figure 7 sensors-25-01190-f007:**
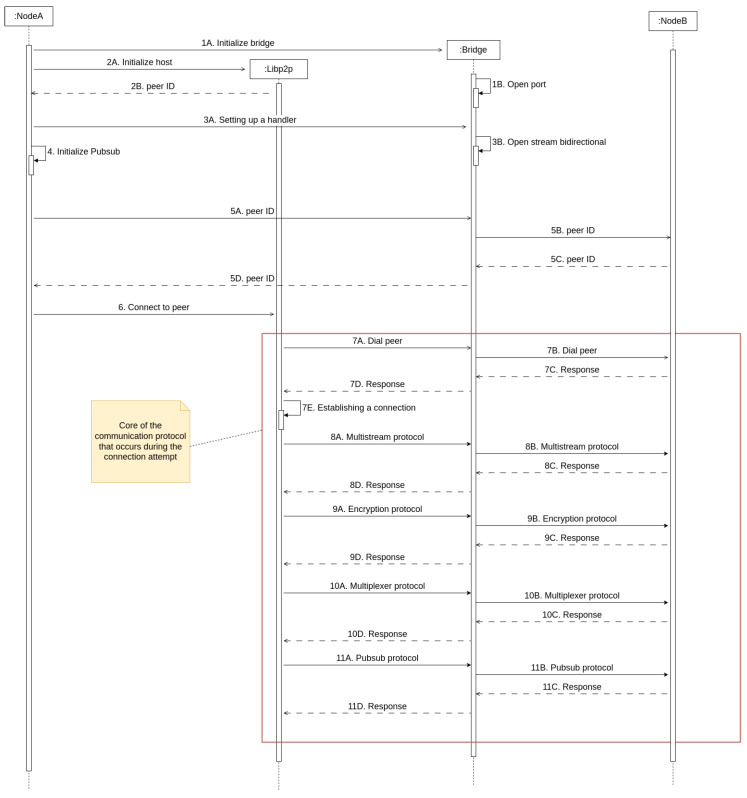
Sequence diagram of the proposed OEC protocol.

**Figure 8 sensors-25-01190-f008:**
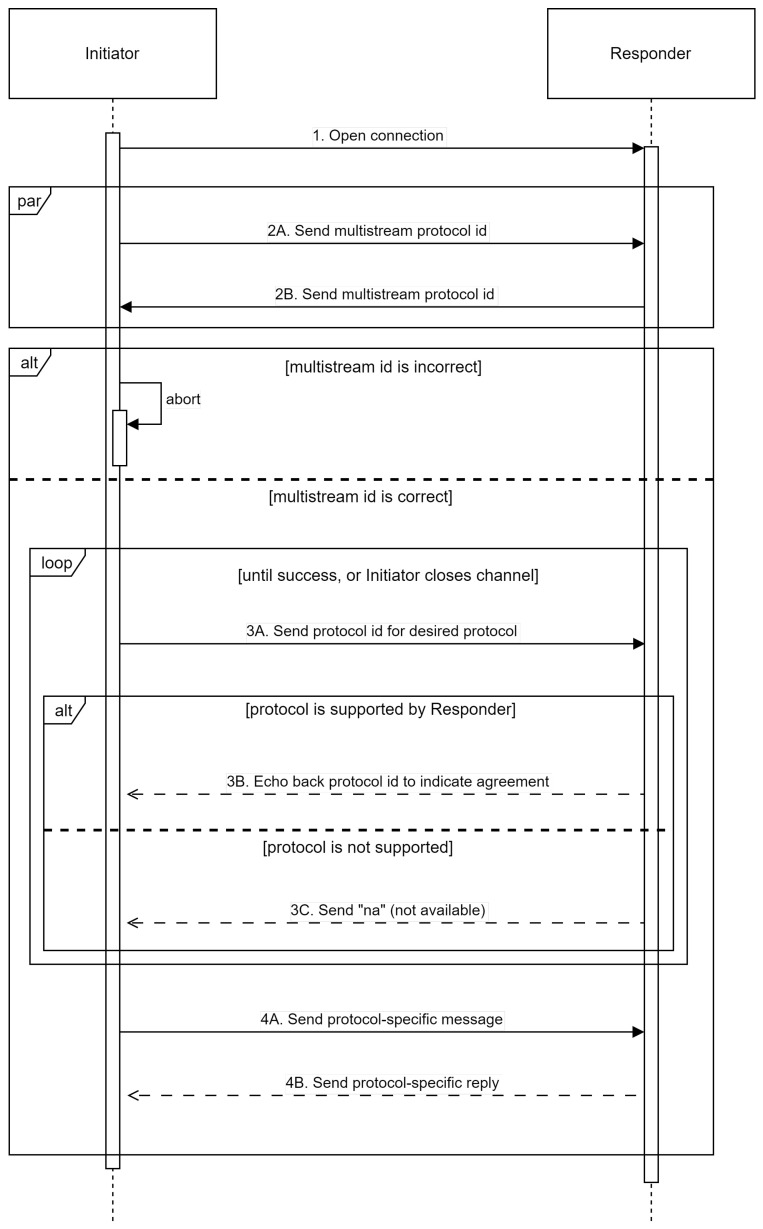
Protocol negotiation flow diagram.

**Figure 9 sensors-25-01190-f009:**
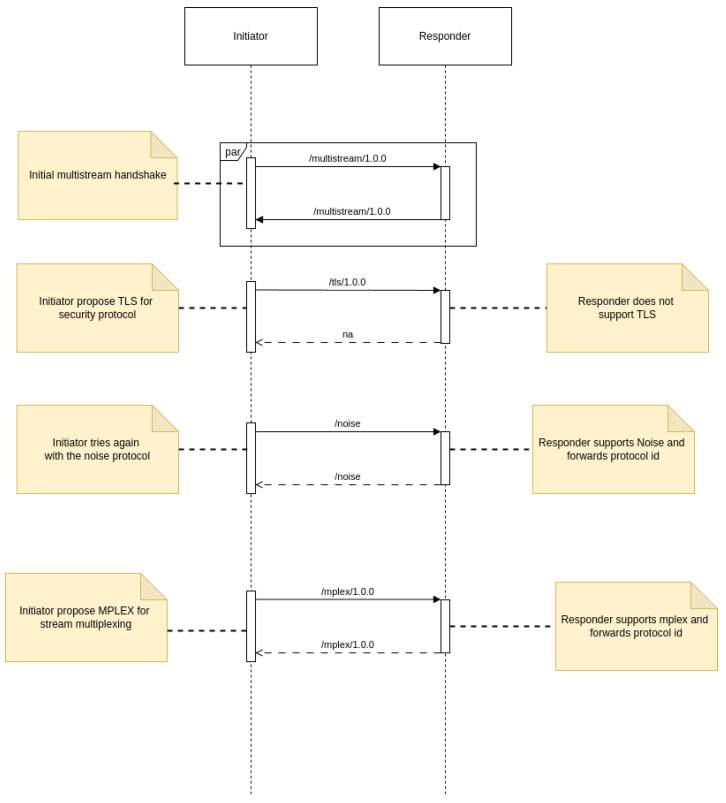
Connection update process sequence diagram.

**Figure 10 sensors-25-01190-f010:**
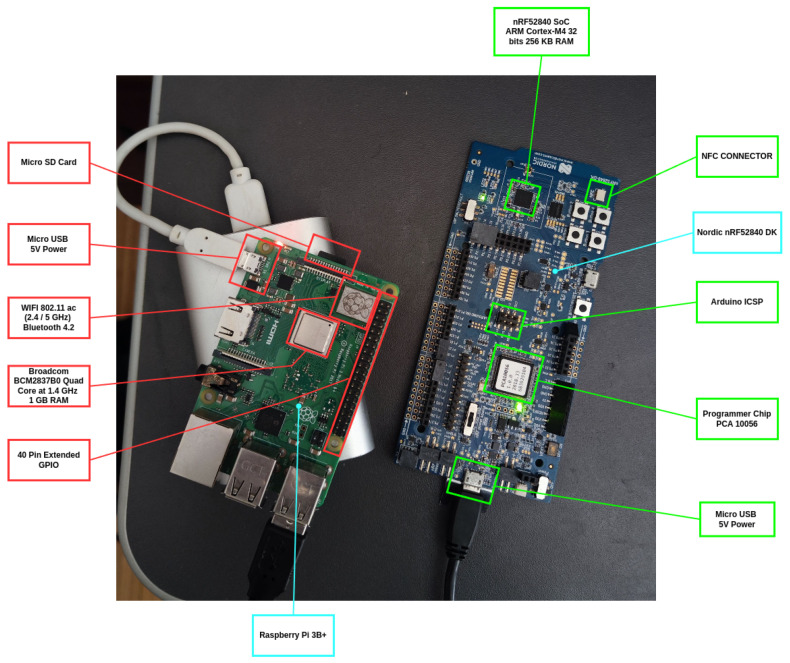
Components of a testbed IoT node.

**Figure 11 sensors-25-01190-f011:**
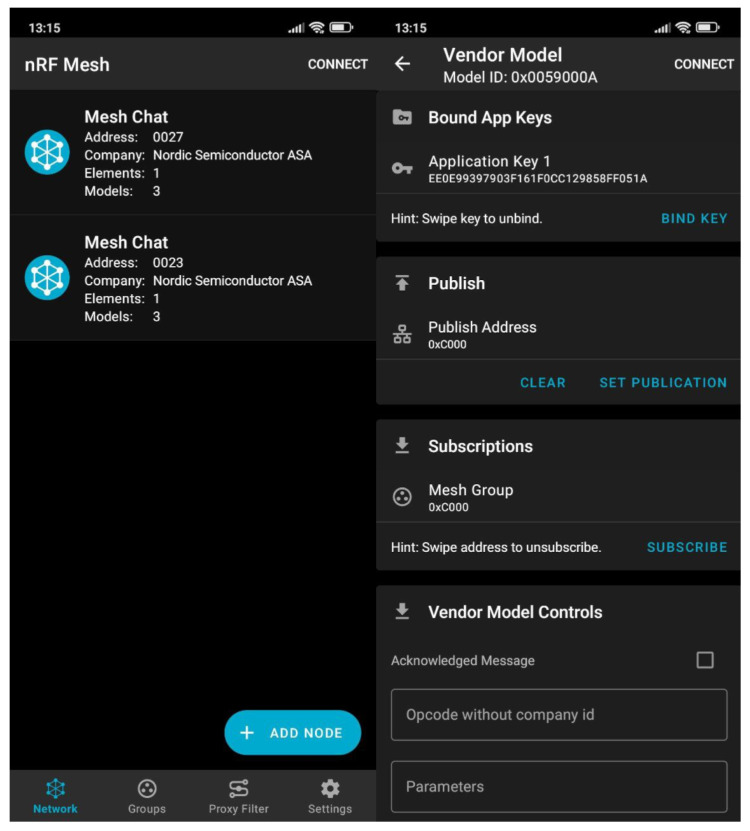
Screenshots of the nRF mesh app.

**Figure 12 sensors-25-01190-f012:**
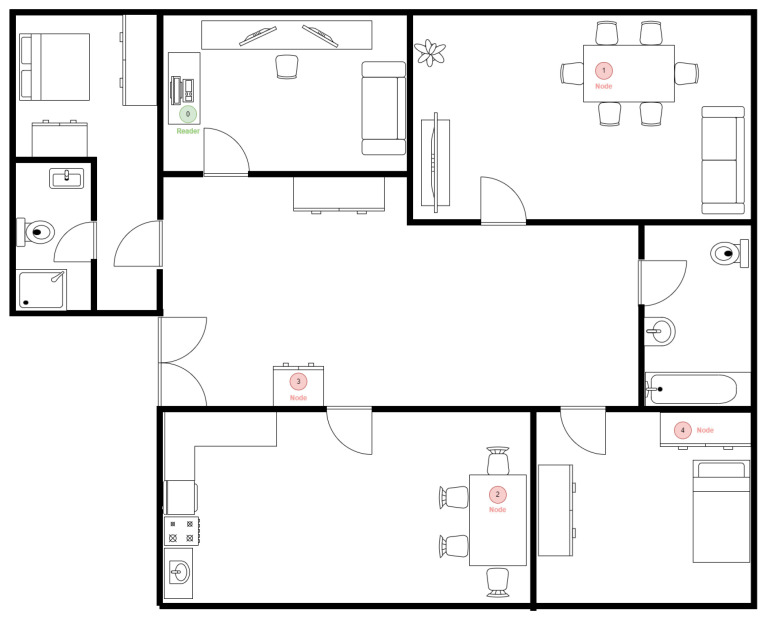
Floor plan of the home automation scenario.

**Figure 13 sensors-25-01190-f013:**
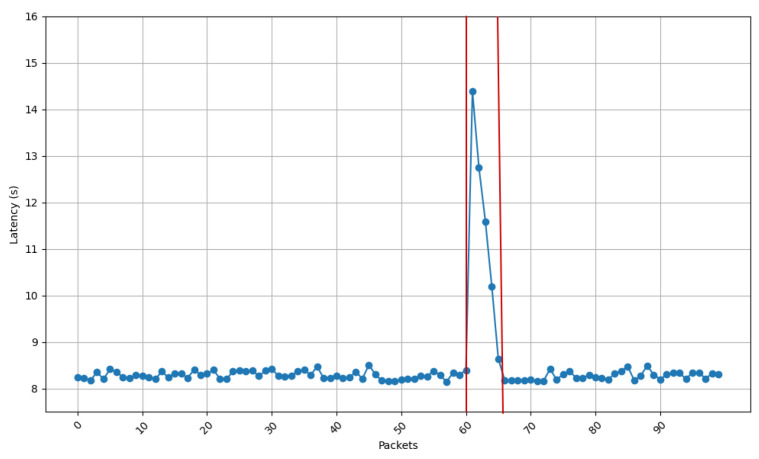
Latency for node at position 1.

**Figure 14 sensors-25-01190-f014:**
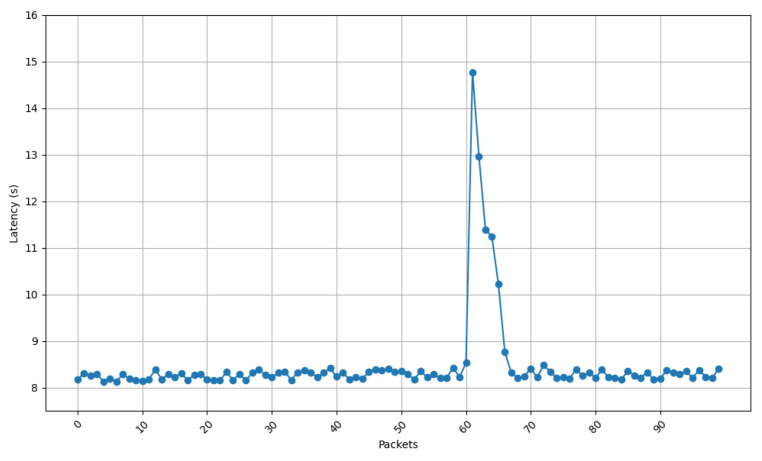
Latency for node at position 3.

**Figure 15 sensors-25-01190-f015:**
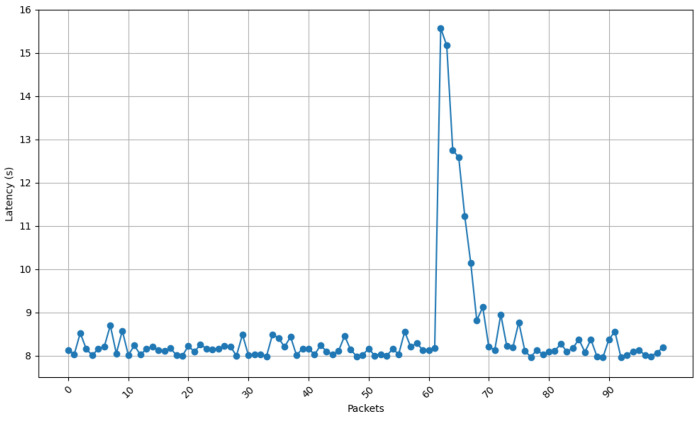
Latency for node at position 2.

**Figure 16 sensors-25-01190-f016:**
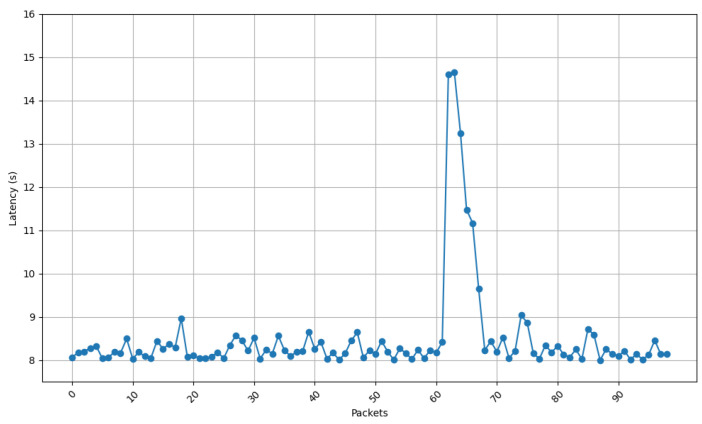
Latency for node at position 4.

**Figure 17 sensors-25-01190-f017:**
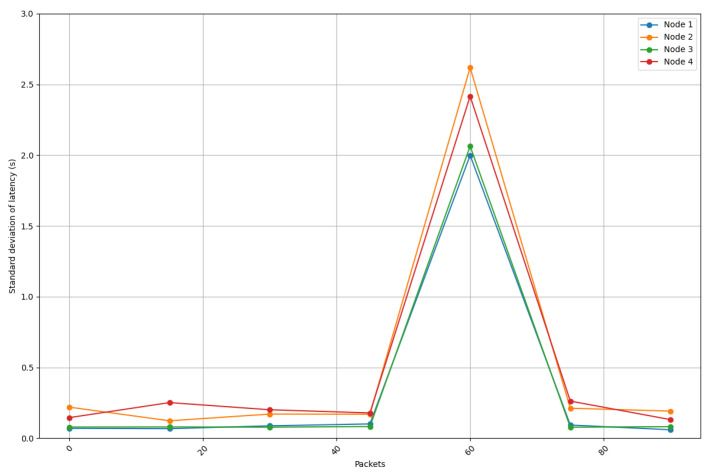
Latency variability (standard deviation) for the four analyzed home scenarios.

**Figure 18 sensors-25-01190-f018:**
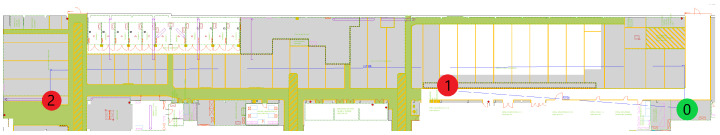
Floor plan of the workshop scenario.

**Figure 19 sensors-25-01190-f019:**
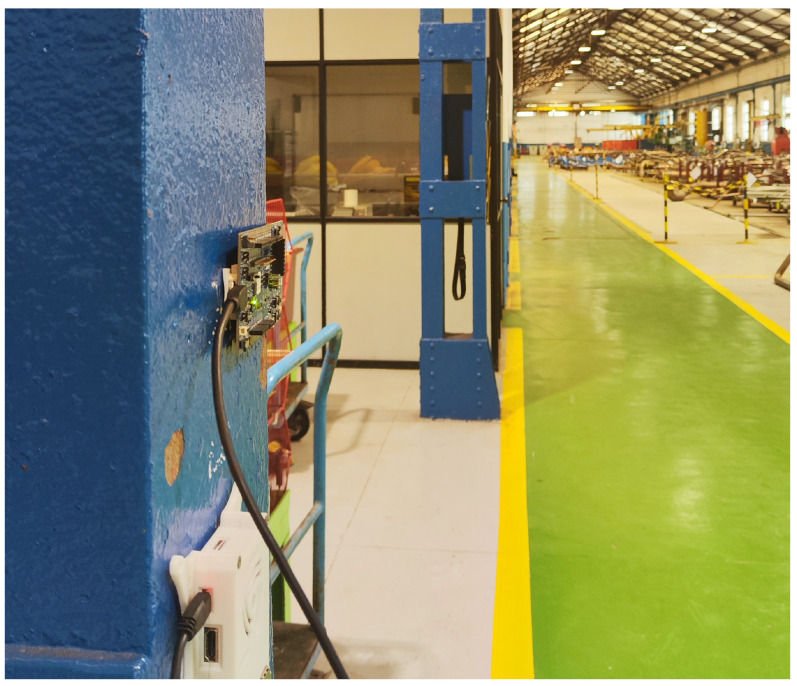
Transmitting node in the workshop.

**Figure 20 sensors-25-01190-f020:**
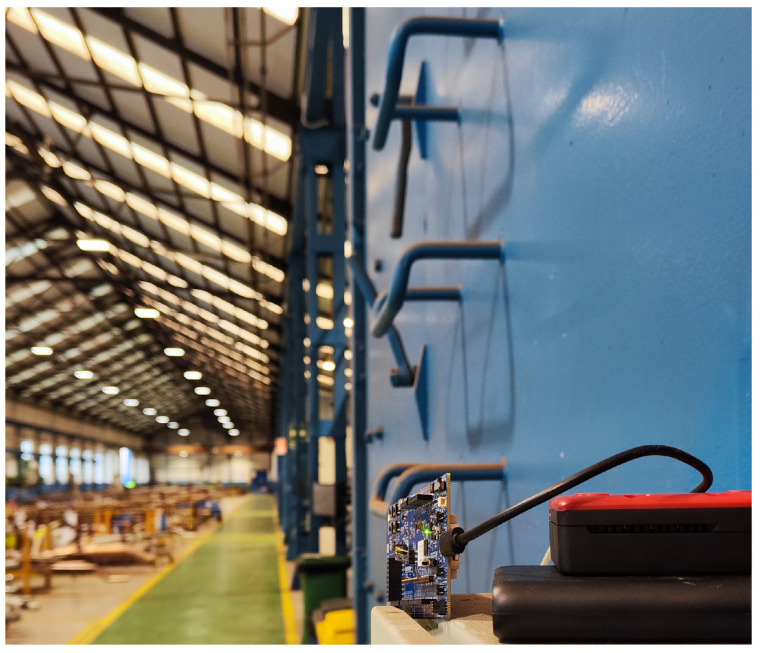
Receiving node in the workshop.

**Figure 21 sensors-25-01190-f021:**
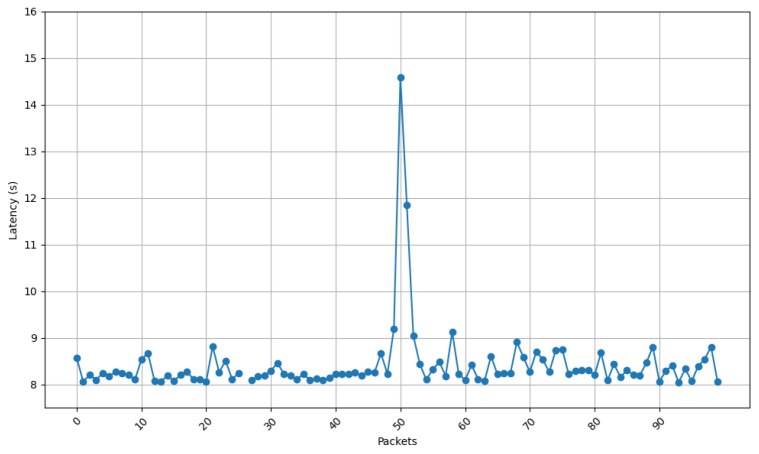
Latency for node at position 1.

**Figure 22 sensors-25-01190-f022:**
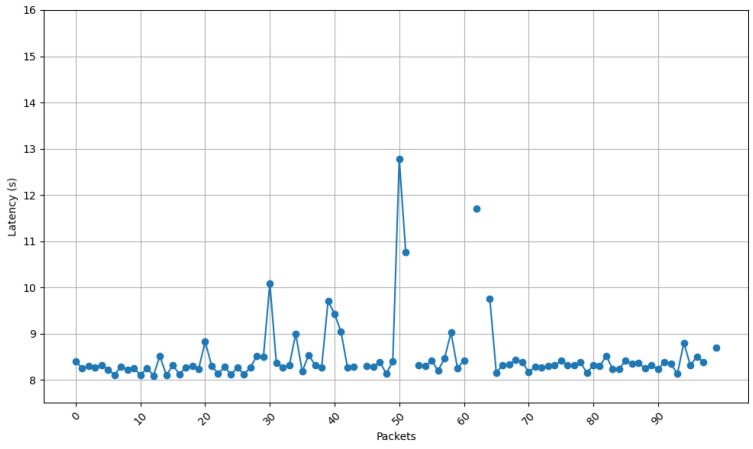
Latency for node at position 2.

**Figure 23 sensors-25-01190-f023:**
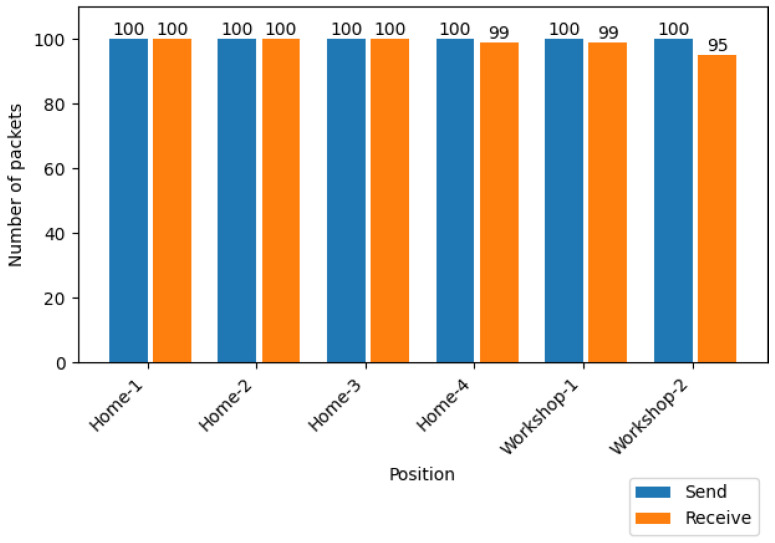
Number of packets successfully sent and received in the six evaluated spots.

**Table 1 sensors-25-01190-t001:** Main characteristics of the most relevant communications technologies with potential application to OEC.

Technology	Standard	Frequency Band	Maximum Range	Data Rate	Topology	Battery Life	Power Efficiency	Scalability	Latency	Cost
RFID	-	HF 3–30 MHz (13.56 MHz), LF 30–300 KHz (125 KHz), UHF 30 MHz–3 GHz	from cm (LF) to tens of meters (UHF)	<640 Kbit/s	-	Long durability	-	-	-	Low
NB-IoT	3GPP	LTE in-band, guard-band (700–900 MHz)	<15 km	<250 Kbit/s	Star	10 years	Very high	Yes	<10 s	High
LTE-M	3GPP	LTE in-band, guard-band (700–900 MHz)	100 km (LTE)	<1 Mbit/s	Star	10+ years	Medium	Yes	10–15 ms	High
LoRa, LoRaWAN	LoRa Alliance	2.4 GHz	2–5 km (urban), 15 km (suburban), 45 km (rural)	0.25–50 Kbit/s	Star on star	8–10 years	Very high	Yes	1–2 s	Low
SigFox	Sigfox	868–902 MHz	3–10 km (urban), 30–50 km (rural)	100 Kbit/s	Star	8–10 years (140 12-byte messages per day)	Very high	Yes	1–30 ms	Medium
Weightless W/N/P	Weightless SIG	License-exempt sub-GHz	up to 15 km	1 Kbit/s–10 Mbit/s (W), 30–100 Kbit/s (N), 200 bit/s–100 Kbit/s (P)	Star	<10 years	Very high	Yes, 10-byte messages (W/P) and up to 20-byte (N)	Low	Low
Ingenu	Ingenu, IEEE 802.15.4k	2.4 GHz	15 km (urban), 80 km (rural)	624 Kbit/s (DL), 156 Kbit/s (UL)	Star, tree	>10 years	Very high	Yes	10–20 s	Medium
Wi-Fi	IEEE 802.11b/ g/n/ac	2.4–5 GHz	<250 m	11 Mbit/s (b), 54 Mbit/s (g), 1 Gbit/s (n/ac)	Point to hub	Days, up to 1 year (AA battery)	Medium	Limited	<50 ms	Low
Bluetooth, BLE	IEEE 802.15.1	2.4 GHz	50–100 m	<24 Mbit/s (v. 4)	Point-to-point, star, mesh	Years on coin cell battery	Very high	Limited	3 ms	Low
Bluetooth 5	-	2.4 GHz	Up to 240 m (ideal conditions)	Up to 2 Mbit/s	Point-to-point, point-to-multipoint, mesh network	Highly efficient	Excellent	Large number of devices in a mesh network	3 ms	Low
ZigBee	ZigBee Alliance	868–915 MHz, 2.4 GHz	<100 m	20−250 Kbit/s	Star, mesh, cluster tree	From months to years	Very high	Yes (up to 65,536 nodes)	15 ms	Low
Ultra Wideband (UWB)	IEEE 802.15.3a, IEEE 802.15.4a, IEEE 802.15.4z	3.1 to 10.6 GHz	< 10 m (depending on the environment)	>110 Mbit/s	Point-to-point, mesh network	Low power consumption (depending on usage)	Very efficient for high-precision applications	Yes	10 ms	Higher than Bluetooth, but cost-effective for specialized use cases

**Table 2 sensors-25-01190-t002:** Internal structure of the protocol data frame.

13 bytes	6 bytes	2 bytes	1–227 bytes	4 bytes	3 bytes
HEADER	DST	START CHAR	DATA	LENGTH	END CHAR

## Data Availability

Data is contained within the article.

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
