# Peer review of "Design, Implementation and Practical Evaluation of an Opportunistic Communications Protocol Based on Bluetooth Mesh and libp2p"

_sensors, 2025, doi:10.3390/s25041190_

Round 1

Reviewer 1 Report

Comments and Suggestions for Authors

Author Response

Comments 1: While the literature review discusses several existing technologies, it would benefit from including more recent advancements in Bluetooth 5 and libp2p. Highlight comparisons with alternative protocols or similar frameworks to establish a more robust foundation for the study. Recently, information freshness and RF-level parameter estimation have been extensively studied, and the following references are advised to be investigated.

[1]  Improving Age of lnformation for Covert Communication with Time-Modulated Arrays.

[2]   Integrated Sensing and Communication with Massive MIMO: A Unified Tensor Approach for Channel and Target Parameter Estimation.

Response 1: Thank you for the recommendation. We agree with the reviewer that the literature review lacked more recent developments, so we have added the references provided by the reviewer. References to the articles on lines 129-132 and 139-144 have been added to section 2.2.

Comments 2: What specific criteria guided the choice of Bluetooth Mesh over alternative technologies like LoRa or Zigbee for this protocol? How does the integration of libp2p address the limitations of Bluetooth Mesh mentioned in the introduction, such as power consumption and message routing complexity?

Response 2: Thank you for the comment. Bluetooth 5 is preferred over other technologies like Wi-Fi, LoRa or Zigbee because it provides a better balance among power consumption, data speed, range and mesh networking capabilities. This makes it particularly ideal for mobile-centric applications, where seamless connectivity with smartphones and wearables, along with energy efficiency, are essential. A summary of the key features of the most relevant communication technologies with potential application in OEC has been added to the new Table 1.

The explanation of why Libp2p was used to mitigate Bluetooth 5 weaknesses was found in the previous version of the article in the section describing Libp2p (Section 4.5). However, to clarify why Libp2p is so important, we decided to expand Section 4.5 by adding details of how Libp2p addresses Bluetooth Mesh limitations (lines 373-388).

Comments 3:  Why was latency chosen as the primary metric, and were there any additional metrics (e.g., throughput, energy consumption) evaluated to provide a more comprehensive performance assessment?

Response 3: We appreciate the reviewer’s insightful comment. Latency was chosen as the key performance indicator because, in the case of IoT scenarios like disaster response, industrial monitoring or asset tracking, the timing of data reception plays an important role in the decision-making process. In opportunistic networks that allow messages to be sent even when there is a temporary disconnection and where the arrangement of the nodes changes frequently, it is crucial to minimize latency so that enough time is available for the message to reach the destination before the connection times out. In order to clarify this issue to the reader, we have added an explanation to section 5 (lines 553-561) of why latency measurement is so important for this kind of protocol.

Moreover, to address the reviewer’s concern appropriately, we have enhanced the evaluation section by adding an additional metric: the Packet Delivery Ratio (PDR). As detailed in Section 5.3.2, we evaluated PDR in both domestic and industrial environments. Thus, PDR complements latency by quantifying reliability in different scenarios. 

Comments 4: Can the observed latency spikes (e.g., at packet 60) be attributed solely to protocol health-check mechanisms, or might other factors (e.g., network congestion) contribute?

Response 4: Thank you for the comment. To address this issue, in the new version of the manuscript (in lines 595-600) it is indicated the reason of the mentioned peak: after checking the connection logs and ruling out possible connectivity failures, it was confirmed that the peak occurs because after a certain number of minutes, the protocol verifies whether the communication is still active, which causes a delay in the remaining packets

Comments 5: How would the protocol perform under scenarios with significantly larger node densities or higher message traffic?

Response 5: We appreciate the reviewer's insightful comment. The experiments shown in the article were on purpose aimed at evaluating the viability of using the system on selected home and industrial scenarios, showing its basic functionality with a reduced minimal P2P environment with only two nodes. Such evaluations were carried out for this article to control all the variables of the experiments and thus accurately show the basic performance of the protocol, such as delivery latency and success rate, without “interference” from other factors like the interactions from multiple concurrent nodes. However, in real-world scenarios, pubsub protocols are designed to operate in distributed networks with hundreds or thousands of nodes, so further research is still necessary in terms of scalability. 

To clarify this issue, we have added a new section 6 (“Key findings”) where the previous justification is provided and where we encourage future researchers to evaluate more complex scenarios. In addition, as a reference, we indicate a paper where a protocol was tested on a large network obtaining a really good performance by combining low latency, high resistance to attacks and self-recovery mechanisms.

Comments 6: Could the proposed protocol integrate emerging technologies like 6G or blockchain to enhance scalability or trustworthiness in IoT networks?

Response 6: We fully agree with the reviewer. We added to the previously mentioned new section 6 (“Key findings”) the fact that the proposed protocol can be adapted and benefit from using other communications and application-level technologies to improve scalability and reliability in IoT networks. Specifically, in lines 669-679, the new version of the manuscript indicates that the protocol is oriented to be agnostic to the physical layer and thus be adapted in future developments to other communications technologies such as 6G or to application-level technologies like blockchain.

Reviewer 2 Report

Comments and Suggestions for Authors

This paper presents a protocol that relies on Bluetooth 5 and libp2p to establish a peer to peer communication. Experiments presented are conducted in static home and industrial environments.

The approach presented has merits. Mainly, by leveraging libp2p, the opportunistic communication scheme can leverage the known advantages of libp2p such as peer discovery, encryption and identity management. 

However, the quality of the work presented can be significantly improved if there are discussions addressing some of the inherent challenges related to libp2p such as adaptability to changes in network topology. 

One of the contributions listed states that "a novel integration of libp2p with Bluetooth 5 in order to enable peer-to-peer (P2P) features like peer discovery and decentralized management, which enhance network flexibility and reliability." However, this claim has not been proven in the work because of the size of the network.  In addition, the dynamic nature of the peer discovery seems to be lacking because all the experiments provided are based on two nodes, in a static environment.

Other minor/easily fixable items:

Figure 3: the OEC block here can be more thoroughly explained in the first paragraph on page 6 

Figures 4 and 5: the blocks inside these figures should be clearer. The text in the Sequence diagram is not readable. 

On Page 9, Section 4.3.2, please consider breaking the following sentence: "After all segments ... are decoded."

Author Response

Comments 1: This paper presents a protocol that relies on Bluetooth 5 and libp2p to establish a peer to peer communication. Experiments presented are conducted in static home and industrial environments.

The approach presented has merits. Mainly, by leveraging libp2p, the opportunistic communication scheme can leverage the known advantages of libp2p such as peer discovery, encryption and identity management. 

However, the quality of the work presented can be significantly improved if there are discussions addressing some of the inherent challenges related to libp2p such as adaptability to changes in network topology. 

Thank you for the comment. To address the concerns of the reviewer, we have added a new Section 4.5.4 by adding libp2p inherent challenges (lines 485-501). One of these challenges is adapting to constant topological changes, along with the bandwidth and energy costs incurred for peer discovery, as well as limitations in routing algorithm efficiencies. Furthermore, it is also noted that libp2p can be resource intensive, affecting its deployment among IoT devices with limited capacity. To address some of these issues, it is suggested that peer discovery be delegated to Bluetooth Mesh and libp2p would take care of overall connectivity and routing, thereby improving system efficiency.

Comments 2: One of the contributions listed states that "a novel integration of libp2p with Bluetooth 5 in order to enable peer-to-peer (P2P) features like peer discovery and decentralized management, which enhance network flexibility and reliability." However, this claim has not been proven in the work because of the size of the network.  In addition, the dynamic nature of the peer discovery seems to be lacking because all the experiments provided are based on two nodes, in a static environment.

We appreciate the reviewer's insightful comment. The experiments shown in the article demonstrated were on purpose aimed at evaluating the viability of using the system on selected static home and industrial scenarios, showing its basic functionality with a reduced minimal P2P environment with only two nodes. Such evaluations were carried out for this article to control all the variables of the experiments and thus accurately show the basic performance of the protocol, such as delivery latency and success rate, without “interference” from other factors like the interactions from multiple concurrent nodes or the dynamic nature of the tested environments. However, in real-world scenarios, pubsub protocols are designed to operate in distributed networks with hundreds or thousands of nodes, which can be mobile, so further research is still necessary in terms of scalability and mobility. 

To clarify this issue, we have added a new section 6 (“Key findings”) where the previous justification is provided and where we encourage future researchers to evaluate more complex scenarios. In addition, as a reference, we indicate a paper where a protocol was tested on a large network obtaining a really good performance by combining low latency, high resistance to attacks and self-recovery mechanisms.

Other minor/easily fixable items:

Comments 3: Figure 3: the OEC block here can be more thoroughly explained in the first paragraph on page 6 

We fully agree with the reviewer. The description for Figure 3 was poor, so we added lines 259-273 in section 4.1, where we explain in more detail what the protocols selected for the implementation of the opportunistic protocol consist of.

Comments 4: Figures 4 and 5: the blocks inside these figures should be clearer. The text in the Sequence diagram is not readable. 

Thank you for the comment. We certainly agree with the reviewer: Figures 4 and 5 have been enhanced to improve its readability.

Comments 5: On Page 9, Section 4.3.2, please consider breaking the following sentence: "After all segments ... are decoded."

Thank you for the comment. The text of such a sentence in Section 4.3.2 was separated for better readability. 

Reviewer 3 Report

Comments and Suggestions for Authors

The paper presents Design, Implementation and Practical Evaluation of an Opportunistic Communications Protocol Based on Bluetooth Mesh and libp2p. However, it does not clarify how will the Opportunistic Network be formed or configured for the proposed communication protocol.

I believe that the topic has some merit, but the paper needs more work to build a clear story of how everything will be constructed. 

Further comments:  

1. There is no clear pictorial depiction of how opportunistic network will actually work in the proposed system! The paper talks about communication sequences in software coding terms, but does not explain exactly how the decentralization will happen and how will the sharing of communication happen in the opportunistic network? 

2. In section 3, the design explanation presents three main layers. It seems stability of OEC Gateway layer plays the key role in sharing information when missing communication happens. If so, that needs to be spelled out consistently in the other sections of the paper. If one Gateway router fails (in Figure 1), how can we ensure information from IoTs connected to that Gateway can be handled by other Gateway routers? A disaster recovery plan just for OEC Gateway layer seems necessary to make the opportunistic network to work properly.

3. If OEC gateway layer is the way to maintain the low-power IoT communications in opportunistic network, overall IoT communication cost may increase. The paper needs some analysis on cost budget.

4. Other than communication latency, not enough measurable contribution was presented in the paper.

5. Some texts in Figure 4 and Figure 5 are not readable.

Author Response

Comments 1: There is no clear pictorial depiction of how opportunistic network will actually work in the proposed system! The paper talks about communication sequences in software coding terms, but does not explain exactly how the decentralization will happen and how will the sharing of communication happen in the opportunistic network? 

Response 1: We fully agree with the reviewer. To tackle this issue, we have added a new Subsection 2.1 where we explain in detail what an opportunistic protocol consists in, referencing previous work that studied how an opportunistic protocol works and what it is used for. To better illustrate how it works, we added Figure 1, which is a generic opportunistic architecture. Based on such a Figure, the new manuscript details in lines 78-110 how the opportunistic process occurs, from information collection to data storage. Moreover, it is explicitly described the previous work that served as the basis for the developments presented in the article. The description includes an explanation on how the sharing of communication happens as follows: (1) the IoT devices deployed in remote areas use Bluetooth 5 to dynamically connect to edge gateways based on Single-Board Computers (SBCs) when these are within their communication range; (2) such gateways act as intermediary nodes that provide local storage and processing services, reducing the dependency on a continuous Internet connection.

Comments 2: In section 3, the design explanation presents three main layers. It seems stability of OEC Gateway layer plays the key role in sharing information when missing communication happens. If so, that needs to be spelled out consistently in the other sections of the paper.

If one Gateway router fails (in Figure 1), how can we ensure information from IoTs connected to that Gateway can be handled by other Gateway routers? A disaster recovery plan just for OEC Gateway layer seems necessary to make the opportunistic network to work properly.

Response 2: Thank you for the comment. The explanation of how the information is handled by the opportunistic gateways is now reflected in Subsection 2.1 (as previously mentioned in answer (1)) and is illustrated through the description of Figure 1. In addition, it can be clarified that the information received by opportunistic gateways is stored in a distributed manner across the rest of the gateways and that in the event that a node loses its connection, this information is not lost and remains stored within the network. Such redundancy allows for securing the information received from devices connected to the Gateways, employing a peer-to-peer (P2P) network to store data on other nearby gateways or to route them to available destinations, thus ensuring service continuity even when a node loses its connectivity and data locally. 

Comments 3: If OEC gateway layer is the way to maintain the low-power IoT communications in opportunistic network, overall IoT communication cost may increase. The paper needs some analysis on cost budget.

Response 3: We appreciate the reviewer’s insightful comment. Although our study has focused more on testing the operational features of the protocol (latency and packet delivery rate (PDR)), we understand the importance of cost budget estimation. As stated in Section 2.1, the main goal of the proposed protocol is to enable connectivity of otherwise non-connected IoT devices by leveraging the deployed OEC gateways. Our approach assumes that these gateways have sufficient resources, such as a stable power supply and higher-enough processing capacity. We have added this clarification in section 3 (lines 209-211).

Comments 4: Other than communication latency, not enough measurable contribution was presented in the paper.

Response 4: We appreciate the reviewer’s insightful comment. To address the reviewer’s concern appropriately, we have enhanced the evaluation section with a comparison of Packet Delivery Ratio (PDR). As detailed in the new Section 5.3.2, we evaluated PDR in both domestic and industrial environments, thus complementing latency by quantifying reliability in different scenarios. 

Regarding why latency was chosen as the primary metric, we chose latency as the key performance indicator because, in the case of IoT uses like disaster response, industrial monitoring or asset tracking, the timing of data reception plays an important role in the decision-making process. In opportunistic networks that allow messages to be sent even when there is temporary disconnection and where the arrangement of the nodes changes frequently, it is crucial to minimize latency so that enough time is available for the message to reach the destination before the connection times out. In order to clarify this issue to the reader, we have added an explanation to section 5 (lines 553-561) of why latency measurement is so important for this kind of protocol.

Comments 5: Some texts in Figure 4 and Figure 5 are not readable.

Response 5: Thank you for the comment. A new version of the figures has been created to enhance their readability.

Round 2

Reviewer 1 Report

Comments and Suggestions for Authors

After reviewing the author's response, the reviewer felt that the manuscript could be published.

Author Response

The authors would like to thank the reviewer for his/her valuable comments, which have certainly helped us to improve the manuscript.

Reviewer 3 Report

Comments and Suggestions for Authors

The paper still does not portray a clear story of how everything in the proposed system will be constructed. Disaster recovery plan for OEC Gateway is not solid. A disaster recovery plan just for OEC Gateway layer seems necessary to make the opportunistic network to work properly. However, the paper has lot of valuable information, experiment data that can be useful. 

Comments on the Quality of English Language

Instead of lot of random explanations, a simple sequential story of how opportunistic network will work in a Disaster situation would make it more readable. 

Author Response

The authors would like to thank the reviewer for his/her valuable comments, which have certainly helped us to improve the manuscript. Please find below our detailed responses to the comments. In order to ease the labor of the reviewers, in the new manuscript we have colored in red the differences with the previous version of the article.
